# Towards an affect intensity regulation hypothesis: Systematic review and meta-analyses of the relationship between affective states and alcohol consumption

**Anna Tovmasyan**[1,2]*, **Rebecca L. Monk**[1,2], **Derek Heim**[1,2]

**1** Department of Psychology, Edge Hill University, Ormskirk, Lancashire, United Kingdom, **2** Liverpool Centre for Alcohol Research, Liverpool, United Kingdom

* tovmasan@edgehill.ac.uk

**Data Availability Statement:** All data files are available on Open Science Framework: https://osf.io/fe9au/.

## Abstract

While self-medication and positive and negative reinforcement models of alcohol use suggest that there is an association between daily affect and alcohol consumption, findings within the academic literature have been inconsistent. This pre-registered systematic review meta-analytically interrogated the results from studies amongst non-clinical populations that examine the relationship between daily affective states and alcohol consumption volume. PRISMA guided searches of PsychINFO, PsycARTICLES, Science Direct, PubMed, SCOPUS, and JSTOR databases were conducted. When both laboratory and field studies were included, meta-analyses with robust variance estimation yielded 53 eligible studies on negative affect (8355 participants, 127 effect sizes) and 35 studies for positive affect (6384 participants, 50 effect sizes). The significant pooled associations between intra-day affect and alcohol consumption were $r = .09$, [.03, .14] for negative affect, and $r = .17$, [.04, .30] for positive affect. A small-to-medium sized effect ($d = .275$, [.11, .44]) of negative affect on daily alcohol consumption volume was found in laboratory studies (14 studies, 1100 participants). While publication bias was suspected, *P*-curve analyses suggested that the results were unlikely to be the product of publication bias and p-hacking alone, and selection model analysis revealed no significant differences in results when publication bias was accounted for. For negative affect, using number of drinks as the measure of alcohol consumption was associated with lower effect sizes. For positive affect, the results demonstrated a decline of this observed effect over time. Overall, findings point towards the possibility of developing an affect intensity regulation theory of alcohol use. Conceptualizing the mood-alcohol nexus in terms of affect intensity regulation may afford a more parsimonious explanation of alcohol consumption rather than viewing the behavior as being shaped by either positive or negative affective states.

**Funding:** This work was funded by Edge Hill University PhD Studentship.

**Competing interests:** The authors have declared that no competing interests exist.

## Introduction

In many Western societies the link between alcohol and mood is deeply rooted, and this cultural knowledge is aptly illustrated by Bukowski's [1, p. 373] assertion that "*If something bad happens you drink in an attempt to forget; if something good happens you drink in order to celebrate; and if nothing happens you drink to make something happen*." Not content with leaving the relationship between mood and alcohol consumption to the writers and philosophers, scientists have, for many years, sought to investigate this association. As a result, a large body of research and theory that seeks to shed light on the extent to which people drink because of their mood has accrued. However, research findings have been mixed and attempts to synthesise this literature meta-analytically have been limited to a study examining negative affect in laboratory settings [2]. This found a small-to-moderate effect of negative mood inductions on alcohol consumption in research that was conducted in contexts that differ markedly from those in which people typically drink. Therefore, the current systematic review and meta-analyses aimed to extend this formative work by focussing on positive as well as negative affect and examining methodically empirical findings garnered from studies carried out in both laboratory and in real world settings.

Both the self-medication hypothesis [3] and the negative reinforcement model of alcohol use [4] theoretically posit a direct association between diminished mental wellbeing and hazardous drinking [5], whereby alcohol is used as a means of improving low mood. While these theoretical models were initially developed to help understand heavy consumption in clinical populations [6,7], they have also been applied to explain excessive drug and alcohol use in non-clinical settings with a view of preventing the development of substance use disorders (e.g., [8–10]). In a similar vein, tension-reduction [11] and stress-response dampening theories [12] construe the experience of negative affect as a risk factor for problematic consumption. The decision to drink (or to exercise restraint), according to these theories, is a product of people's affective experiences, and alcohol consumption is understood temporally [12–14] as an outcome of their preceding (negative) mood.

Empirical findings examining the mood-alcohol nexus have, however, been varied. While some laboratory studies indicate that negative affect is associated with increased alcohol-related attentional bias [15,16], others suggest that it is not related to alcohol consumption [17–19]. Similarly, some real-time studies find no association between negative affect and unplanned heavy drinking [20], while others suggest that negative affect is inversely related to drinking onset and further intoxication [21–25]. The existing literature therefore provides mixed support for negative affect regulation theories of alcohol use.

In a similar way, there are inconsistent findings with regards to the positive reinforcement theory of alcohol use [26], according to which people drink alcohol to enhance the positive emotions they are experiencing. As such, while real-time studies indicate that positive mood is associated with increased drinking likelihood and breath alcohol concentration later that day [21,23–25,27], a study using questionnaire design found that having difficulties with regulating positive emotions was linked with drug but not alcohol misuse [28]. Studies conducted in the laboratory also produced mixed findings: while Stein et al. found that positive mood induction increases consumption [29], VandeVeen et al. found no such effect [30]. Given the variation in support for both negative and positive reinforcement theoretical models, formal interrogation of discrepant findings derived from both laboratory and field settings is required to shed light on the somewhat elusive association between affective states and alcohol consumption. It is therefore necessary to meta-analytically examine whether inconsistent findings may be due to power limitations of individual studies.

In addition to overcoming power concerns by combining effect sizes of studies, theoretical and methodological differences between investigations need to be examined systematically to

help clarify inconsistencies in this body of work. The first reason for mixed findings may relate to how affect is conceptualised and measured. Theoretically, there are two perspectives on affective state: while some research operationalises affect as a singular concept and measures this on a continuum that is anchored between positive and negative affect (e.g., [31,32]), in other work current mood is treated as a unidimensional construct in which different affective states are unable to overlap simultaneously (e.g., [33,34]). This has methodological implications as the former conceptualisation of affect necessitates the use of measures that assess mood on a continuous scale, with scope for variability in valence and arousal (e.g., mood circumplex, [35]; UMACL, [36]; affect grid, [37]), while the latter perspective typically uses separate assessments of intensity of negative and positive affect (e.g., PANAS, [38]; VAMS, [39]). These theoretical and methodological considerations are important as they also have consequences for how study findings are interpreted. If affect is understood and measured as a continuous construct, evidence suggesting that positive affect increases consumption (in line with the positive reinforcement model) would contradict the negative reinforcement model. On the other hand, if positive and negative affect are assessed as discrete entities, then evidence for one theory would not necessarily contradict the other and it may be beneficial to combine these approaches into a more parsimonious model. When taking stock of this literature it is therefore essential to consider how measurement choices and theoretical conceptualisations of affect may impact results.

A second potential methodological reason for divergent findings in this research area centres on whether the studies examine distinct emotions or look at overall levels of affect. An important debate in the literature concerns whether emotions ought to be conceptualised along dimensions of valence and arousal [40,41] or as discrete entities [42,43]. In the alcohol literature, the first approach, where researchers measure mood scores (e.g., PANAS, [38]) as an average of various adjectives related to either negative or positive emotions (e.g., [27,44–46]), has been more commonly utilised. Yet, an alternative approach, which adopts a discrete model of emotions, analyses each affect item individually. O'Donnell et al., for example, examined how stress and irritation impact consumption [47], while Dvorak and Simons [9] as well as Shadur et al. [33] looked at how anxiety and sadness affect drinking likelihood, and Rohsenow et al. [48] examined how anxiety, anger, and depression influence the number of alcohol units consumed. Considering that emotions differ in terms of arousal [35,49,50] and physiology [43], it is possible that the widely adopted methodology of averaging affective states may have inadvertently contributed to a homogenisation of different facets of emotions. It therefore needs to be examined whether the process of collapsing distinct emotions into positive and negative affect scores may have led researchers to miss the nuanced ways in which these emotions shape alcohol consumption differentially, or whether combining them into a single score for negative and positive affect is appropriate.

The third methodological variation between studies that may systematically impact results relates to their design. On the one hand, daily diary and ecological momentary assessment (EMA)/experience sampling methods (ESM) studies have enabled researchers to minimise retrospection bias [51,52] and to examine the behaviour in question in naturalistic settings. Here, participants are instructed to record in structured ways events/feelings that occurred during the day. However, such studies occur in uncontrolled environments, and it is therefore possible that findings may be impacted by extraneous factors that are not captured by the research methods used in these studies. Laboratory studies, on the other hand, typically utilise ad-libitum drinking paradigms, where participants can consume as much or as little alcohol as they wish. While having a controlled environment is advantageous, participants may feel obliged to drink alcohol [53,54] or may not be offered their typical beverage of choice (alcoholic or non-alcoholic, [55,56]) and therefore might not accurately reflect real-world drinking behaviours.

The potential for study design to moderate the association between affect and alcohol consumption therefore needs to be considered meta-analytically.

Accordingly, the present pre-registered systematic review and meta-analyses aimed to synthesise findings on the impact of affective states on alcohol consumption in non-clinical populations by addressing these gaps, while accounting for potential sources of variability. Specifically, following the self-medication hypothesis and tension-reduction theory, which postulate that increases in negative affect predict substance use within a short timeframe [13,14], we examined the impact of affective states on same day consumption. In consideration of the suggestion that negative and positive affect may be distinct experiences [57,58] separate models were used in the analyses. Laboratory studies examining negative affect were analyzed separately prior to the main analysis (which examined both laboratory and field studies) to test for any causal association. For both negative and positive affect models, alcohol measure (e.g., number of drinks or units), affect conceptualization (i.e., whether studies treated negative and positive affective states as a continuum or separate entities and whether studies looked at distinct emotions or averaged them) and study design (i.e., laboratory or field research) were examined systematically as possible methodological moderators of the affect-alcohol relationship. A series of exploratory analyses of other variables (year of publication, country, study quality) was also undertaken. The results of two meta-analyses (on negative affect and on positive affect) were then compared to establish whether negative and positive affect are differentially associated with alcohol consumption volume.

## Methods

### Operational definitions

Alcohol consumption is defined as ingesting any beverage containing ethanol. Mood and emotions are distinct but related constructs in that the former tend to be more stable and 'flat', while the latter are construed as more vivid and quick [59]. However, studies sometimes use these terms interchangeably. While it is possible that mood and emotions have different effects on alcohol consumption volume over longer periods of time, the current focus was on the effects of within a shorter timeframe, where the distinction between mood and emotions is arguably less important. Therefore, to account for differences in the terminology, the terms 'affect', and 'affective state' are used in this review as umbrella terms for the experience of mood, emotion, or feeling. We use the term 'field studies' for real-time studies, diary studies or studies using telephone interviews.

### Eligibility criteria

The literature search was primarily conducted by the lead author. To avoid missing data, the second author conducted a comparative title search using the same criteria to ensure the incorporation of any studies which may have been overlooked in the original review. Full-text papers of any titles and abstracts that were considered relevant were obtained where possible. The relevance of each study was assessed according to the following inclusion criteria (pre-registered on Open Science Framework: https://osf.io/8bngj): peer-reviewed papers; grey literature; focus on the general human population (non-clinical sample); looking at affective states on the day of and prior to consumption; looking at consumption volume during the day (e.g., amount consumed in millilitres, numbers of drinks); papers in English or Russian. The exclusion criteria were as follows: reviews, books, posters, and editorials; literature examining clinical samples (individuals with alcohol use disorders or any other clinical disorder).

Both studies that measured affect as a continuum (i.e., where positive and negative affect are at polar ends of the same assessment spectrum), or as separate entities were included in the

review. Furthermore, studies that examined mean levels of affect (i.e., average negative or positive affect) as well as affect facets (i.e., specific emotions, e.g., stress, anger, or happiness) were included. To account for varied methods of assessment, both laboratory and field studies were included. Methodological differences (treating affect as continuum or separate entities, examining mean levels of affect or distinct emotions, alcohol measure used, laboratory or field studies) were included in analysis as moderators.

## Literature review

A comprehensive search was conducted of the following databases: PsychINFO, PsycARTICLES, Science Direct, PubMed, SCOPUS, JSTOR using Preferred Reporting Items for Systematic Review and Meta-Analyses (PRISMA, [60]) and American Psychological Association's Meta-Analysis Reporting Standards (MARS; [61,62]) methodologies. The following commands were used for searching: ("alcohol " OR "drinking behavi*r") AND ("mood" OR "emotions" OR "feelings" OR "affective states") NOT "disorders". The search was conducted on 2nd March 2020. For PsychINFO, after the filters 'empirical study' and 'quantitative study' were applied, the search yielded 8285 articles for screening. For PsychARTICLES, when the same filters were applied, the search yielded three articles. For Science Direct, as wildcards "*" were not supported, the search terms were ("alcohol " OR "drinking behavior" OR "drinking behaviour") AND ("mood" OR "emotions" OR "feelings" OR "affective states") NOT "disorders". After the filter 'research articles' was applied, the search yielded 2327 articles. For PubMed (3189 citations), SCOPUS (1201 citations), and JSTOR (367 citations), no filters were applied. The citations were loaded to RefWorks software, and the duplicates were removed. Bibliographies from relevant reviews and book chapters, as well as articles that fit the inclusion criteria, were manually searched for additional citations. To ensure that all relevant literature published at the time was covered, a supplementary search was conducted on 29th January 2021, which yielded 3 additional references.

To obtain grey literature, Google Scholar and Open Science Framework were searched. We also contacted the labs that conduct studies on the topics of affect and alcohol consumption. However, only one grey literature study (a study from our own lab) was included in the review, as other available studies did not fit the inclusion criteria (e.g., did not examine same day affect and alcohol consumption).

## Quality assessment and data extraction

Study quality was assessed using standard criteria [63], with papers screened by two independent reviewers (Cohen's Kappa = .71). Each paper was rated on the following criteria (each criterion assessed on a scale from 0 to 2): justification of research question, justification of study deigns, appropriate method of study selection, robustness of the measures, sample size justification, appropriateness of analytic methods, estimation of variance, control for confounds, results being reported in sufficient details, and conclusion being supported by results, with a maximum quality score of 22. Scores of 1–10 were considered to be poor quality, those that scored 11–15 were deemed to be of moderate quality, and studies with scores of 16–20 were classified as being of good quality, with manuscripts scoring 21–22 bring considered to be excellent quality. None of the studies were judged to be of poor quality (and hence none were excluded based on this), while there were 20 studies of moderate quality, 36 that were good quality, and two that were deemed to be of excellent quality.

Following the quality assessment, relevant data were extracted from each study (see Table 1 for full summary). For subset of laboratory studies on negative affect, Cohen's *d* statistics was extracted (by calculating the mean difference between the two groups, and then dividing

**Table 1. Characteristics of included studies.**

| Number | Authors | Year | Country | Method | Sample size(s) | Sample type | Sample gender(s) | Sample age(s) | Positive or negative | Affect measure | Looked at distinct emotions (Yes/No) | Alcohol measure | Effect size (r) | Relevant key findings |
|---|---|---|---|---|---|---|---|---|---|---|---|---|---|---|
| 1 | Austin, Notebaert, Wiers, Salemink, & MacLeod [80] | 2020 | Netherlands | Experimental, ad libitum taste test | 39 | General population | 24 women | M = 24.82, SD = 7.49, range = 18–59 | Negative | VAMS [39] | No | Ad libitum (grams of beverage consumed) | r = .72 | Beer consumption was higher in the negative affect induction condition. |
| 2 | Cyders, Zapolski, Combs, Settles, Fillmore, & Smith [81] | 2010 | USA | Experimental study, ad libitum taste test | 33 | Undergraduate psychology students | 14 women | M = 22.27, SD = 2.36, range not reported | Positive | PANAS [38] | No | Ad libitum (millilitres of beverage consumed) | r = .40 | Participants drank more in the positive mood than in the neutral mood condition. |
| 3 | de Castro [82] | 1990 | USA | Diary study | 96 | General population | 63 women | M = 32.9, SD not reported, range = 21–54 | Both, as a continuum | 7-point Likert scale (elated—depressed, anxious—calm scales) | No | Record everything you drink | r (elation-depression scale) = .33; r (anxious-calm scale) = .17 | The degree of elation, not depression, was related to the amount of alcohol ingested. On the calm-anxiety scale, people were mostly drinking when they felt 'neutral'. |
| 4 | Dinc & Cooper [83] | 2015 | UK | Experimental study, ad libitum taste test; mood induction | 106 | Undergraduate psychology students | 60 women | M = 23.915, SD = 6.73, range not reported | Positive | UMACL [36] | No | Ad libitum (millilitres of beverage consumed) | r = .43 | People consumed more alcohol when in a positive mood compared to a neutral mood. |
| 5 | Duif, Thewissen, Wouters, Lechner, & Jacobs [44] | 2019 | Netherlands | Experience Sampling Method | 162 | General population | 109 women | M = 36.07, SD = 9.27, range = 20–50 | Both | PANAS [38] | No | Number of drinks | Within persons: r (daily negative affect) = .01; r (momentary negative affect) = −.03; r (daily positive affect) = .03; r (momentary positive affect) = .01; r (daily negative affect) = .01, r (momentary negative affect) = .03; r (daily positive affect) = .01; r (momentary positive affect) = .01. Between persons: r (daily negative affect) = .04. | Negative affect was not associated with the amount consumed. Higher levels of positive affect were associated with more consumption. |

*(Continued)*

**Table 1.** (Continued)

| Number | Authors | Year | Country | Method | Sample size(s) | Sample type | Sample gender(s) | Sample age(s) | Positive or negative | Affect measure | Looked at distinct emotions (Yes/No) | Alcohol measure | Effect size (r) | Relevant key findings |
|---|---|---|---|---|---|---|---|---|---|---|---|---|---|---|
| 6 | Gabel, Noel, Keane & Lisman [84] | 1980 | USA | Experimental, ad libitum taste test | 18 | Undergraduate psychology students | Men only | M & SD not reported, range = 18–22 | Both | Self-report; basal skin conductance; heart rate | Yes: sexual arousal, fear, neutral | Ad libitum (millilitres of beverage consumed) | r (negative) = .11; r (positive) = .65 | Participants in the sexual arousal condition drank more than in fear or neutral conditions. |
| 7 | Gautreau, Sherry, Battista, Goldstein, & Stewart [85] | 2015 | Canada | Experience Sampling Method | 143 | Frequent drinkers | 105 women | M = 20.78, SD = 3.36, range not reported | Positive | PANAS [38] and mood circumplex [86] | No | Number of drinks | r (low arousal) = .03; r (high arousal) = .26 | High-arousal positive mood was associated with higher number of drinks than low-arousal positive mood. |
| 8 | Liu, Wang, Zhan, & Shi [87] | 2009 | China | Telephone Interviews | 37 | General population | 5 women | M = 31, SD = 8.01, range not reported | Negative | Work stress checklist [88] | Yes—stress | Number of units | r = .16 | Stress was associated with increased alcohol consumption. |
| 9 | Mohr, Arpin & McCabe [89] | 2015 | USA | Experience Sampling Method | 47 | Moderate-to-heavy drinkers | 23 women | M = 36, SD = 16.98, range not reported | Both | PANAS [38] and mood circumplex [86] | No | Number of drinks | r (negative) .01; r (positive) = .04 | Affect variability, not mean levels of affect, was associated with elevated consumption. |
| 10 | Mohr, Brannan, Wendt, Jacobs, Wright, & Wang [90] | 2013 | USA | Experience Sampling Method | 49 | Moderate-to-heavy drinkers | 24 women | M = 36, SD = 17.32, range not reported | Both | PANAS [38] and mood circumplex [86] | No | Number of drinks | r (positive) = -.16 | Participants drank more following increases in both positive and negative mood. |
| 11 | Monk, Qureshi, & Heim [32] | 2020 | UK | Experience Sampling Method | 69 | General population | 42 women | M = 21.47, SD = 4.47, range = 18–36 | Both, as a continuum | How would you rate your current mood (on a scale from 0 to 5)? | No | Number of drinks | r (negative) = -.31; r (positive) = .31 | Feeling unhappy prior to the commencement of drinking was a significant predictor of drinking larger quantities of alcohol in the following drinking session. |
| 12 | O'Donnel et al. [47] | 2019 | Australia | Experience Sampling Method | 83 | General population | 63 women | M = 21.42, SD = 3.09, range = 18–30 | Both (happy, relaxed, irritated, stressed) | Happy, relaxed, irritated, stressed, on a 6-point scale | Yes (happy, relaxed, irritated, stressed) | Number of drinks | r (happy) = .09; r (relaxed) = .10; r (stressed) = -.09; r (irritated) = -.04 | Affect was not related to levels of consumption. |

(Continued)

Table 1. (Continued)

| Number | Authors | Year | Country | Method | Sample size(s) | Sample type | Sample gender(s) | Sample age(s) | Positive or negative | Affect measure | Looked at distinct emotions (Yes/No) | Alcohol measure | Effect size (r) | Relevant key findings |
|--------|---------|------|---------|--------|----------------|-------------|------------------|---------------|----------------------|----------------|--------------------------------------|-----------------|-----------------|-----------------------|
| 13 | Peacock, Cash, Bruno, & Ferguson [91] | 2015 | Australia | Experience Sampling Method | 53 | General population | 22 women | M = 28.2, SD = 11.2, range = 18–60 | Both, as a continuum | Visual-analogue Mood Scales [35] | No | Number of drinks | r (negative) = -.19; r (positive) = .19 | Higher positive affect was associated with increased alcohol consumption. |
| 14 | Pihl & Yankofsky [92] | 1979 | Canada | Experimental study, ad libitum taste test; mood induction | 40 | General population | Men only | M = 20.05, SD not reported, range = 18–27 | Negative | MAACL [93] | Yes (depression and anxiety) | Ad libitum (total amount of pure alcohol consumed) | r = .29 | Less alcohol was consumed by participants who experienced higher depression and anxiety prior to consumption. |
| 15 | Richardson, Hoene, & Rigatti [45] | 2020 | USA | Diary study | 222 | University students | 149 women | M = 20.12, SD = 2.58, range not reported | Both | PANAS [38] | No | Number of drinks | r (negative) = .11; r (positive) = .17 | At low levels of positive affect, individuals higher in self-critical perfectionism reported higher levels of drinking to cope than those lower in self-critical perfectionism. Individuals were also more likely to drink to cope with high negative affect compared to low negative affect. |
| 16 | Rohsenow [94] | 1982 | USA | Experimental study, ad libitum taste test; mood induction | 60 | Undergraduate students | Men only | M = 23, SD not reported, range = 21–32 | Negative | MAACL [93] | Yes (anxiety) | Ad libitum (amount consumed in mls, average sip size, total number of sips) | r = .31 | Those feeling more anxious took fewer sips of alcohol. |
| 17 | Simons, Gaher, Oliver, Bush, & Palmer [95] | 2005 | USA | Experience Sampling Method | 56 | Moderate drinkers | 30 women | M = 21.5, SD = .57, range = 21–23 | Both | PANAS [38] and mood circumplex [86] | No | Number of drinks | r (negative) = .03; r (positive) = .16 | Both negative and positive affect were associated with greater consumption volume. |

(Continued)

**Table 1.** (Continued)

| Number | Authors | Year | Country | Method | Sample size(s) | Sample type | Sample gender(s) | Sample age(s) | Positive or negative | Affect measure | Looked at distinct emotions (Yes/No) | Alcohol measure | Effect size (r) | Relevant key findings |
|---|---|---|---|---|---|---|---|---|---|---|---|---|---|---|
| 18 | Simons, Wills, & Neal [96] | 2014 | USA | Experience Sampling Method | 274 | Moderate-to-heavy drinkers | 153 women | M = 19.88, SD = 1.37, range = 18–27 | Both | PANAS-X [97] and mood circumplex model [86] | No | Number of drinks + transdermal alcohol monitoring | r (negative) = .11; r (positive) = -13 | Daily negative affect was directly associated with higher consumption on drinking days. |
| 19 | Stasiewicz & Lisman [98] | 1989 | USA | Experimental, ad libitum taste test; mood induction | 32 | Men in risk for future child abuse | Men only | M = 20.6, SD & range not reported | Negative | Blood pressure and heart rate | Yes (aversion, arousal, distress) | Ad libitum (millilitres of beverage consumed) | r = .50 | Higher aversion, arousal, and distress were associated with higher consumption. |
| 20 | Stevenson, Dvorak, Kramer, Peterson, Dunn, Leary, & Pinto [99] | 2019 | USA | Experience Sampling Method | 101 | College students | 66 women | M = 20.93, SD = 2.89, range = 18–29 | Both | PANAS [38] | Yes (depression, anxiety) | Number of drinks | r (anxious) = .08; r (depressed) = -.01; r (positive) = -.02 | Intending to drink to enhance one's mood was associated with increased consumption volume. |
| 21 | Sutker, Libet, Allain, & Randall [100] | 1983 | USA | Diary study | 32 | General population | 21 women | Not reported | Negative | MAACL [93] | Yes (anxiety, depression, hostility) | Number of drinks | r (anxiety) = .07; r (depression) = -.10; r (hostility) = -.02 | Negative affect was not associated with consumption. |
| 22 | Swendsen, Tennen, Carney, Affleck, Willard, & Hromi [14] | 2000 | France | Experience Sampling Method | 100 | Frequent drinkers | 55 women | M = 22.9, SD = 4.6, range not reported | Both | Mood Circumplex [86] | Yes (active, peppy, happy, relaxed, quiet, bored, sad, nervous) | Type & Quantity of beverage | r (active) = .11; r (peppy) = −.05; r (happy) = .21; r (relaxed) = .12; r (quiet) = .32; r (bored) = .06; r (sad) = .01; r (nervous) = .30 | Happy and nervous affective states were associated with increased consumption, feeling quiet was associated with decreased consumption. |
| 23 | Wardell, Read, Curtin, & Merrill [101] | 2012 | USA | Experimental, ad libitum taste test; mood induction | 146 | College students | 67 women | M = 21.45, SD = .73, range = 21–24 | Both, as a continuum | Affect Grid [37] | No | Ad libitum (oz of beverage consumed, blood alcohol concentration) | r (negative) = -.07; r (positive) = .07 | Mood was not associated with consumption. |
| 24 | Mohr et al. [102] | 2005 | USA | Experience Sampling Method | 122 | Undergraduate students | 69 women | M = 18.9, SD = 1.16, range not reported | Both | PANAS [38] and mood circumplex [86] | No | Number and volume of drinks | r (negative) = .08; r (positive) = .07 | Both positive and negative mood were positively associated with consumption volume. |

*(Continued)*

**Table 1.** (Continued)

| Number | Authors | Year | Country | Method | Sample size(s) | Sample type | Sample gender(s) | Sample age(s) | Positive or negative | Affect measure | Looked at distinct emotions (Yes/No) | Alcohol measure | Effect size (r) | Relevant key findings |
|---|---|---|---|---|---|---|---|---|---|---|---|---|---|---|
| 25 | Hussong, Galloway, & Feagans [103] | 2005 | USA | Experience Sampling Method | 72 | College students | 36 women | M = 18.10, SD not reported, range not reported | Both | PANAS [38] | Yes (fear, hostility, attentiveness, sadness, shyness) | Number of drinks | r (fear) = -.08; r (hostility) = .17; r (attentiveness) = -.10; r (sadness) = .18; r (shyness) = -.05 | Affect interacted with drinking motives to predict consumption: those high in drinking-to-cope motives drank less on days in which they experienced greater sadness. When experiencing moderate to high levels of fear and shyness, individuals high in drinking-to-cope motives were more likely to drink. For those low in coping motivations, fear and shyness did not predict daily drinking |
| 26 | Todd, Armeli, Tennen, Carney, & Affleck [104] | 2003 | USA | Diary study | 83 | Community sample | 44 women | M = 37.2, SD = 6.65, range not reported | Negative | Perceived stress scale [105] and mood circumplex [86] | Yes (anger, boredom, loneliness, nervousness, sadness) | Number, size, and proof of drinks | Study 1. r (angry) = .05; r (bored) = -.03; r (lonely) = -.10; r (nervous) = -.06; r (sad) = -.05. Study 2. r (angry) = 0; r (bored) = .02; r (lonely) = .05; r (nervous) = -.06; r (sad) = -.06 | Associations between stress/ negative affect and drinking outcome variable tend to be near zero for individuals with high drinking-to-cope scores and negative for individuals with low drinking-to-cope scores. |
| 27 | Carney, Armeli, Tennen, Affleck, & O'Neil [106] | 2000 | USA | Diary study | 83 | Community sample | 44 women | M = 37.15, SD = 6.65, range = 26.01–50.76 | Negative | Perceived Stress Scale [105] | Yes (stress) | Number, size, and proof of drinks | r = .27 | Perceived stress was associated with increased consumption. |

(*Continued*)

**Table 1.** (Continued)

| Number | Authors | Year | Country | Method | Sample size(s) | Sample type | Sample gender(s) | Sample age(s) | Positive or negative | Affect measure | Looked at distinct emotions (Yes/No) | Alcohol measure | Effect size (r) | Relevant key findings |
|---|---|---|---|---|---|---|---|---|---|---|---|---|---|---|
| 28 | Armeli, Carney, Tennen, Affleck, & O'Neil [107] | 2000 | USA | Diary study | 88 | Moderate drinkers | 48 women | M = 37.81, SD = 6.92, range not reported | Negative | Modified version of the Assessment of Daily Experience [108] | Yes (stress) | Number of drinks | r = -.23 | Men who more strongly anticipated positive outcomes or a sense of carelessness from drinking drank relatively more on stressful days compared with low-stress days. Men who anticipated greater impairment from drinking drank relatively less on stressful days. These effects did not hold for women. |
| 29 | Dvorak, Pearson, & Day [109] | 2014 | USA | Experience Sampling Method | 74 | University students | 43 women | M = 21.30, SD = 2.07, range = 18–29 | Both | How ___ are you feeling right now (on a scale from 1 to 11)? | No | Number of drinks | r (negative) = .17; r (positive) = -.14 | Negative daytime mood was associated with increased consumption, positive daytime mood was not associated with consumption. |
| 30 | Thomas, Merrill, von Hofe, & Magid [110] | 2014 | USA | Experimental, ad libitum taste test; mood induction | 112 | Frequent drinkers | 52 women | M = 27.0, SD = 5.16, range not reported | Negative | Subjective units of distress scale; heart rate; mean arterial pressure; and salivary cortisol. | Yes (stress) | Ad libitum (millilitres of beverage consumed, latency to first sip of beer, average sip size, median latency between sips) | r = .10 | The stressor did not result in greater consumption of alcohol. |
| 31 | Grant, Stewart, & Mohr [111] | 2009 | Canada | Experience Sampling Method | 146 | College students | 113 women | M, SD, & range not reported | Negative | How ___ did you feel today (on a scale from 0 to 4)? | Yes (depression and anxiety) | Number of drinks | r (depressed) = -.07; r (anxious) = -.03 | Daily depressed mood did not trigger subsequent evening alcohol consumption and daily anxious mood was protective against subsequent evening drinking. |

(*Continued*)

**Table 1.** (Continued)

| Number | Authors | Year | Country | Method | Sample size(s) | Sample type | Sample gender(s) | Sample age(s) | Positive or negative | Affect measure | Looked at distinct emotions (Yes/No) | Alcohol measure | Effect size (r) | Relevant key findings |
|---|---|---|---|---|---|---|---|---|---|---|---|---|---|---|
| 32 | Steptoe & Wardle [112] | 1999 | UK | Diary study | 79 | Nurses and teachers | 45 women | M = 39.75, SD = 9.95, range not reported | Both | POMS [113] | Yes (anxiety) | Number of units | r (negative) = .58; r (positive) = .58 | Consumption of alcohol tended to be greater on days on which participants reported more positive and less anxious mood. |
| 33 | Hull & Young [114] | 1983 | USA | Experimental, ad libitum taste test; mood induction | 120 | Frequent drinkers | Men only | Over 21, M, SD & range not reported | Negative | MAACL [93] | Yes (anxiety, hostility, depression) | Ad libitum (ounces of beverage consumed) | In high self-conscious subjects: r (anxious) = .44; r (hostile) = .10, r (depressed) = .09. In low self-conscious subjects r (anxious) = .09, r (hostile) = .10, r (depressed) = .09 | Negative mood was related to consumption volume in high self-conscious but not low self-conscious participants. |
| 34 | O'Hara, Armeli, & Tennen [115] | 2014 | USA | Diary study | 1636 | College students | 867 women | M = 19.2, SD & range not reported | Both | Items from Mood Circumplex [86] and PANAS-X [97] | Yes (sadness, anxiety) | Number of drinks | r (anxiety) = -.06; r (anger) = .00; r (sadness) = -.20; r (positive mood) = .02 | Anxiety, anger, and positive mood were positively related to the number of drinks consumed. |
| 35 | Todd et al. [116] | 2005 | USA | Experience Sampling Method | 98 | Community sample | 49 women | M = 43.5, SD & range not reported | Both | Single-item mood measure: How ___ did you feel (on a scale from 0 to 4)? | Yes (peppy, happy, relaxed, bored, sad, nervous, angry, lonely, disappointed) | Number, size, and proof of drinks | r (peppy) = .01; r (happy) = .05; r (relaxed) = .14; r (positive mood) = .09; r (bored) = .13; r (sad) = .07; r (nervous) = -.01; r (angry) = .05; r (lonely) = .06; r (disappointed) = .00; r (negative moods) = .05 | There was no significant association between mood and alcohol consumption. |

(Continued)

**Table 1.** (Continued)

| Number | Authors | Year | Country | Method | Sample size(s) | Sample type | Sample gender(s) | Sample age(s) | Positive or negative | Affect measure | Looked at distinct emotions (Yes/No) | Alcohol measure | Effect size (r) | Relevant key findings |
|---|---|---|---|---|---|---|---|---|---|---|---|---|---|---|
| 36 | Todd, Armeli, & Tennen [117] | 2009 | USA | Experience Sampling Method | 97 | Community sample | 48 women | M = 43.5, SD & range not reported | Both | Mood Circumplex [86] | Yes (angry, bored, disappointed, lonely, nervous, sad) | Number, size, and proof of drinks | r (angry) = .02; r (bored) = .11; r (disappointed) = -.01; r (lonely) = .04; r (nervous) = -.02; r (sad) = .07; r (negative mood) = .04; r (positive mood) = .08 | Affective state was not associated with consumption. |
| 37 | Collins et al. [118] | 1998 | USA | Experience Sampling Method | 37 | Heavy drinkers | 15 women | M = 35.92, SD & range not reported | Both | Mood Circumplex [86] | No | Number of drinks | r (negative) = .97; r (positive) = .97 | Positive but not negative mood predicted excessive drinking. |
| 38 | Ehrenberg, Armeli, Howland, & Tennen [119] | 2016 | USA | Experience Sampling Method | 722 | College students | 391 women | M = 19.24, SD & range not reported | Both | PANAS [38] and mood circumplex [86] | No | Number of drinks | r (negative) = .06; r (positive) = .01 | Consumption level was unrelated to negative affect and positively related to positive affect. |
| 39 | Higgins & Marlatt [120] | 1975 | USA | Experimental, ad libitum taste test | 64 | Undergraduate psychology students | Men only | M & SD not reported, range = 18–26 | Negative | MAACL [93] | Yes (fear of evaluation) | Ad libitum (ounces of beverage consumed and amount of pure alcohol consumed) | r = .35 | Participants expecting to be evaluated drank significantly more alcohol than low-fear control participants. |
| 40 | Holroyd [121] | 1978 | USA | Experimental, ad libitum taste test | 60 | Undergraduate students | Men only | Over 18, M, SD, & range not reported | Negative | State Anxiety Scale | Yes (social anxiety) | Ad libitum (numbers of bottles of beer opened, blood alcohol concentration) | r = .21 | Socially anxious participants and those who received negative social evaluation drank less alcohol. |
| 41 | Dvorak, Pearson, Sargent, Stevenson, & Mfon [122] | 2016 | USA | Experience Sampling Method | 74 | University students | 43 women | M = 21.30, SD & range not reported | Both | PANAS-X [97] and mood circumplex [86] | No | Number of drinks | r (negative) = .10; r (positive) = .09 | Higher positive mood and mood instability were associated with increased consumption. |

(Continued)

**Table 1.** (Continued)

| Number | Authors | Year | Country | Method | Sample size(s) | Sample type | Sample gender(s) | Sample age(s) | Positive or negative | Affect measure | Looked at distinct emotions (Yes/No) | Alcohol measure | Effect size (r) | Relevant key findings |
|---|---|---|---|---|---|---|---|---|---|---|---|---|---|---|
| 42 | Armeli et al. [123] | 2007 | USA | Experience Sampling Method | 98 | Heavy drinkers | 49 women | M = 43.5, SD & reported | Both | PANAS [38] and mood circumplex [86] | Yes (stress) | Number of drinks | r (negative) = .06; r (positive) = .06 | Higher levels of stress and negative affect interacted with individual differences factors to predict increased consumption. |
| 43 | Corcoran & Parker [124] | 1991 | USA | Experimental study, ad libitum taste test; mood induction | 69 | Undergraduate students | 25 women | M, SD, & range not reported | Negative | Personal Evaluation Form | Yes (stress) | Ad libitum (amount of beverage consumed in ounces) | r = .17 | Stress was not associated with consumption. |
| 44 | Kidorf & Lang [125] | 1999 | USA | Experimental, ad libitum taste test; mood induction; within-subject | 84 | Undergraduate students | 42 women | M = 22.5, SD & range not reported | Negative | MAACL [93] | Yes (stress) | Ad libitum (amount of pure alcohol consumed) | r = .08 | Stress was positively related to consumption. |
| 45 | Tucker, Vuchinich, Sobell, & Maisto [126] | 1980 | USA | Experimental study, ad libitum taste test; mood induction | 43 | Heavy social drinkers | Men only | M & SD not reported, range = 18–26 | Negative | How anxious did you feel (on a scale from 1 to 7)? | Yes (stress) | Ad libitum (milliliters of alcohol consumed) | r = −.70 | Stress was positively related to consumption. |
| 46 | McGrath, Jones, & Field [127] | 2016 | UK | Experimental study, ad libitum taste test; mood induction | 100 | Heavy social drinkers | 52 women | M = 20.86; SD = 3.93, range not reported | Negative | POMS [112] | Yes (stress) | Ad libitum (milliliters of beverage consumed) | r = .21 | Stress was positively related to consumption. |
| 47 | Magrys & Olmstead [128] | 2015 | Canada | Experimental study, ad libitum taste test; mood induction | 75 | Undergraduate students | 40 women | M = 20.12, SD & range not reported | Negative | STAI [129] | Yes (stress) | Ad libitum (number of standard drinks consumed, level of intoxication, blood alcohol level) | r = .09 | Stress was positively related to consumption. |
| 48 | Aldridge-Gerry et al. [130] | 2011 | USA | Experience Sampling Method | 365 | College students | 252 women | M = 20.1, SD & range not reported | Negative | Describe the most stressful event that happened that day and rate how stressful it was. | Yes (stress) | Number of drinks | r = -.05 | Stress was negatively related to consumption. |

*(Continued)*

**Table 1.** (Continued)

| Number | Authors | Year | Country | Method | Sample size(s) | Sample type | Sample gender(s) | Sample age(s) | Positive or negative | Affect measure | Looked at distinct emotions (Yes/No) | Alcohol measure | Effect size (r) | Relevant key findings |
|---|---|---|---|---|---|---|---|---|---|---|---|---|---|---|
| 49 | Emery & Simons [131] | 2020 | USA | Experience Sampling Method | 92 | Moderate-to-heavy drinkers, undergraduate students | 58 women | M = 20.17, SD & range not reported | Both | PANAS-X [97] and mood circumplex [86] | No | Number of drinks | r (negative) = .10; r (positive) = .15 | Positive affect was positively associated with consumption. Negative affect was not associated with consumption. |
| 50 | Hamilton, Armeli, & Tennen [132] | 2020 | USA | Diary study, 2 waves | 906 | College students | 489 women | M = 19.18, SD = 1.26 = > M = 24.56, SD = 1.33, ranges not reported | Both | PANAS [38] and Mood Circumplex [86] | No | Number of standard drinks | Wave 1: r (social positive) = .01; r (social negative) = -.06; r (solitary positive) = -.03; r (solitary negative) = .11. Wave 2: r (social positive) = .01; r (social negative) = .00; r (solitary positive) = -.07; r (solitary negative) = -.04 | Whereas daytime positive affect predicted greater social consumption, it was also related to lower solitary alcohol consumption among college students who were low in state social drinking motives. |
| 51 | Tovmasyan, Monk, Bunting, Qureshi, & Heim | Under revision | UK | Experience Sampling Method | 79 | General population | 49 women | M = 29.31, SD = 9.70, range = 20–63 | Both | PANAS [38] | Yes (all items from PANAS, [38]) | Number of drinks | r (between day positive) = -.05; r (between day negative) = .01; r (within day positive) = -.03; r (within day negative) = .03 | Being more enthusiastic and less alert was associated with drinking onset, being ashamed was associated with higher number of drinks following drinking onset, feeling strong and interested was associated with decreased drinking volume. |
| 52 | Stamates, Linden-Carmichael, Preonas, & Lau-Barraco [46] | 2019 | USA | Experience Sampling Method | 24 | Adult drinkers | 14 women | M = 23.83, SD = 1.83, range not reported | Both | PANAS [38] | No | Number of standardised drinks | r (negative) = .41; r (positive) = -.09 | Higher negative affect was inversely related to number of drinks consumed. |

(*Continued*)

**Table 1.** (Continued)

| Number | Authors | Year | Country | Method | Sample size(s) | Sample type | Sample gender(s) | Sample age(s) | Positive or negative | Affect measure | Looked at distinct emotions (Yes/No) | Alcohol measure | Effect size (r) | Relevant key findings |
|---|---|---|---|---|---|---|---|---|---|---|---|---|---|---|
| 53 | Mohr, Branman, Mohr, Armeli, & Tennen [133] | 2008 | USA | Diary study | 118 | College students | 67 women | M = 18.9, SD = 1.16, range not reported | Both | Combination of PANAS [38] and mood circumplex [86] | Yes (angry, sad, bored, nervous, ashamed, hostile, guilty, jittery, dejected) | Number of drinks | Drinking at home: r (angry) = .21; r (sad) = .04; r (bored) = .15; r (nervous) = –.05; r (ashamed) = .21; r (hostile) = .27; r (guilty) = .12; r (jittery) = .07, r (dejected) = .12, r (positive mood) = .05. Drinking away: r (angry) = .04, r (sad) = –.07; r (bored) = .05; r (nervous) = –.04; r (ashamed) = .12; r (hostile) = .13; r (guilty) = .05; r (jittery) = –.02; r (dejected) = .01; r (positive mood) = .09 | Both positive and negative were associated with higher consumption volume. |

(*Continued*)

**Table 1.** (Continued)

| Number | Authors | Year | Country | Method | Sample size(s) | Sample type | Sample gender(s) | Sample age(s) | Positive or negative | Affect measure | Looked at distinct emotions (Yes/No) | Alcohol measure | Effect size (r) | Relevant key findings |
|---|---|---|---|---|---|---|---|---|---|---|---|---|---|---|
| 54 | Schroder & Perrine [134] | 2007 | USA | Interactive Voice Response | 173 | General population | 81 women | M = 42.3, SD = 11.9, range = 21–74 | Both | Rate stress, anger, sadness, happiness, quality of the day (the best day I had, the worst day I had) on an 11-point scale | Yes (stress, anger, sadness, happiness) | Number of standard drinks | Between-subject: r (stress) = -.08; r (anger) = -.05; r (sadness) = -.04; r (happiness) = -.10; r (negative emotions) = -.06; r (positive emotions) = -.12. Within-subject: r (sadness) = -.02; r (anger) = -.01; r (stress) = -.03; r (happiness) = .10, r (negative emotions) = -.02; r (positive emotions) = .12 | Among women, those with higher average levels of sadness, anger, and stress reported higher levels of alcohol consumption; among men, those with higher negative mood ratings reported significantly less alcohol consumption. When not separated by gender, on both within- and between-participant levels, correlations of mood and drinking did not differ significantly from zero. |
| 55 | Waddell, Sher, & Piasecki [135] | 2021 | USA | Experience Sampling Method | 403 | General population | 202 women | M = 23.3, SD = 7.2, range = 18–70 | Negative | 5-point Likert scale of distressed and sad | Yes (distressed, sad) | Number of drinks | r = .07 | Negative affect did not predict consumption directly but did so through alcohol craving. |
| 56 | Larsen, Engels, Granic, & Huizink [136] | 2013 | Netherlands | Experimental study, ad libitum taste test; mood induction | 106 | University students | Men only | M = 21.37, SD = 2.32, range = 18–27 | Negative | Physiological Arousal Questionnaire (PAQ; [137]) | Yes (stress) | Ad libitum (centilitres alcohol consumed | r = .14 | There was no difference in alcohol consumed between stress and no-stress conditions. |
| 57 | Sacco et al. [138] | 2015 | USA | Telephone interviews | 71 | Continuing care retirement community | 45 women | M = over 80, SD and range not reported | Both | PANAS-S [139] | No | Number of standard drinks | r (negative) = -.08; r (positive) = .07 | No temporal relationship between negative and positive affect and amount consumed. |

(*Continued*)

**Table 1.** (Continued)

| Number | Authors | Year | Country | Method | Sample size(s) | Sample type | Sample gender(s) | Sample age(s) | Positive or negative | Affect measure | Looked at distinct emotions (Yes/No) | Alcohol measure | Effect size (r) | Relevant key findings |
|---|---|---|---|---|---|---|---|---|---|---|---|---|---|---|
| 58 | Lindgren et al. [140] | 2018 | | Experimental study, ad libitum taste test; mood induction | 149 | University students | 71 women | M = 21.55, SD = .68, range = 21–25 | Both | 6-Item Brief Affect Measure [141] | No | Ad libitum (amount in mls) | r (negative) = .07, r (positive) = -.07 | Implicit alcohol excite associations were more negatively associated with drinking in negative mood condition and more positively associated with drinking in positive/neutral mood condition. |

results by pooled standard deviation, or converted from F value, [64]), along with corresponding standard error. Correlation coefficients (*r*) were extracted either from correlation tables (N = 26 for negative affect, N = 19 for positive affect), obtained from the authors (N = 2 for negative affect, N = 2 for positive affect), or converted from available statistics (N = 24 for negative affect, N = 13 for positive affect), such as standardised beta weights (using formula provided by Peterson and Brown [65]), converted from *d* obtained from unstandardised beta weight and pooled standard deviation [66] or from sample size, means, and F-test [67]. When necessary, we changed the direction of correlation coefficient to ensure that each effect size reflected the relation between higher levels of affect and higher consumption volume. The standard error for each effect size was calculated using the following formula: $SE(r) = \sqrt{1-r^2}/N-2$ (following [68]), while the variance was obtained by squaring the standard error.

## Meta-analyses

**Analytical strategy.**   Prior to the main analysis, laboratory studies which provided Cohen's *d* and its standard error (or other statistics from which these numbers could be calculated) were analysed separately. This was only done for studies examining the impact of negative affect on alcohol consumption (n = 14), as there were only two eligible laboratory studies on positive affect for this analysis. Random effects model was fitted in *R Studio* [69] version 1.4.1106 using *metagen* function of the *meta* [70] package.

Pearson's *r* correlation coefficients were used as the effect size for the main meta-analyses, with generic inverse-variance pooling to combine correlations from different studies into one pooled correlation estimate. As Pearson's *r* is not normally distributed, effect sizes were first converted to Fisher's *z* using the following formula: ($z = \frac{1}{2}$ ln ((1+r)/(1-r)). After the analysis, the coefficients were converted back to Pearson's *r* (following [68]).

Some data sets provided multiple correlations between constructs of interest (e.g., the correlation between sadness and consumption and anger and consumption, or both within- and between-person associations; N = 25 for negative affect, N = 16 for positive affect). Given that including more than one effect size from a study violates the assumption of independence, we used the robust variance estimation (RVE; [71]) method to control for dependencies between effect sizes. Because correlations between the effect sizes reported within each study were not known, we assumed a Spearman's rho ($\rho$) of .80 [71]. We also performed a series of sensitivity analyses by testing different values of $\rho$ in intervals of .10. This did not affect inferences about effect sizes; therefore, these results are not reported in the paper. Correlated effects model with small-sample corrections was fitted in *R Studio* [69] using *robumeta* [72] package. Heterogeneity was assessed using $I^2$ and $\tau^2$ statistics.

To assess potential publication bias, we conducted the Egger's test [73], which was performed by regressing effect estimates against their standard errors. If the slope for the regression line is significant, that would suggest publication bias. Additionally, selection model analysis was performed using JASP ([74], following Bartoš et al. [75]). P-curve analysis was also performed in the online app (http://p-curve.com/) to assess potential p-hacking [76].

Several categorical moderators were examined: study quality, country, study design (laboratory vs field), alcohol consumption measure (e.g., number of millilitres consumed during the day, number of drinks consumed during the day), whether studies examined distinct emotions or averaged them, and whether study considered affective state to be a continuum or not. The effects of categorical moderators (i.e., country and study design) were assessed using meta-regression approach, as suggested by Harrer et al. [77]. Additionally, year of publication was a continuous moderator, which was examined using *metatest* [78] package. While examining the differences between the sample types was initially planned, it was deemed inappropriate

due to inconsistent reporting—for example, many studies included anyone who was not diagnosed with alcohol use disorder, while others only included heavy drinkers. Thus, there is a potential overlap between sample types of different studies. As mean AUDIT scores were not reported consistently, we decided not to include sample type as a moderator.

The results of meta-analyses of negative and positive affect were then compared using Cohen's $q$ statistic [79] by imputing the obtained correlations to online calculator (https://www.psychometrica.de/effect_size.html, Lenhard & Lenhard. [Unpublished]). This method transforms correlation coefficients to $z$ scores and then subtracts them.

Data and R scripts for the analyses are available on the Open Science Framework: https://osf.io/fe9au/files/.

## Results

### Quantity of research available

Electronic and hand search identified 15372 articles, which, once duplicates were removed, left 2472 unique citations to be screened for inclusion (Fig 1). Their titles and abstracts were assessed for their relevance to the review, resulting in 22 potential articles being retained. The full texts of all but three studies were obtained. After applying exclusion criteria for the remaining full-text papers, nine articles were excluded; the most common reason for exclusion was that the studies did not look at affective state on the day and prior to consumption. After that, full texts of eligible articles were screened to obtain additional citations. This resulted in screening 264 additional articles. All but 22 were retrieved. After applying exclusion criteria for the remaining full-text papers, 206 were excluded; the most common reason for exclusion was that studies did not examine the variables of interest. Additionally, one study from our laboratory which is currently in preparation was included. Following the supplementary search,

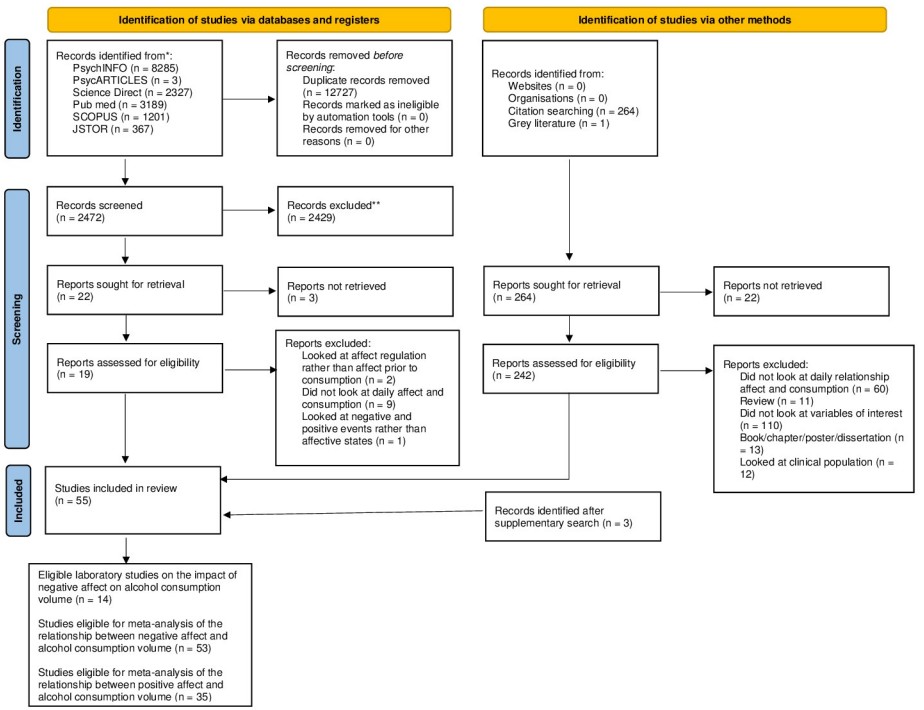

**Fig 1. Flowchart of study selection process.**

two additional articles were included. Overall, 58 studies were eligible for systematic review. Fifty-five studies were eligible for meta-analysis on negative affect, however, two did not allow for effect size extraction, leaving 53 studies. For the meta-analysis on positive affect, 35 studies were eligible and included in analysis. The PRISMA flow diagram summarises the included studies for both negative and positive affect (see Fig 1).

Included studies were published between 1975 and 2021 and were conducted in various countries: Australia (n = 2), Canada (n = 4), China (n = 1), France (n = 1), Netherlands (n = 3), the USA (n = 42), and the UK (n = 5). Most of the studies were either conducted in the laboratory and looked ad libitum consumption, or utilized experienced sampling method (EMA), or similar (diary study, telephone interviews throughout several days). Study characteristics and effect sizes are provided in Table 1.

## Association between daily negative affect and volume of alcohol consumed

**Analysis of laboratory studies.** Fourteen studies (1100 participants) were included in this meta-analysis. Analysis revealed a significant post-mood induction increase in amount of alcohol consumed by participants that was small-to-medium effect size, $d$ = .28, 95% CI [.11, .44], t = 3.351, $p$ = .004. Heterogeneity between the studies was significant, $I^2$ = 47.7%, Q (13) = 24.87, $p$ = .024.

**Main analysis.** A total of 127 effect sizes were extracted from 53 studies (8355 participants). Correlations between negative affect and consumption ranged from -.33 to .70. The pooled correlation coefficient for our data was $r$ = .09, 95% CI [.03, .14], $t$ (48.5) = 3.32, $p$ = .002. As per $I^2$ and $\tau^2$ indexes, $I^2$ = 70.02%, whereas $\tau^2$ = .02.

**Publication bias.** According to Egger's test, there was a publication bias ($t$ = 3.53, $p <$ .01). Selection model also demonstrated publication bias, $\chi^2$ (3) = 25.67, p < .001. After adjusting for publication bias, the relationship between negative affect and drinking volume was still positive and significant, $r$ = .17, 95% CI [.07, .26], $p$ = .004.

*P*-curve analysis indicated that evidential value is present, and that evidential value is not absent or inadequate (see Fig 2). This means that *p*-curve estimates that there is a "true" effect size underlying finding, and that the results are unlikely to be the product of publication bias and *p*-hacking alone. When correcting for selective reporting, the power of tests included in the meta-analysis was 69% (see Fig 3).

**Meta-regression.** Several moderators were examined: year of publication, country, study design (laboratory vs field), study quality, alcohol measure, whether studies examined distinct emotions or averaged them, and whether study considered affective state to be a continuum or not. Alcohol measure was a significant moderator, as studies that looked at number of drinks as an outcome produced significantly lower effect sizes than studies that used other measures. On the other hand, analysis demonstrated that studies that looked at number of units as an outcome produced higher effect sizes, however, since the degrees of freedom were lower than four, this estimate could not be trusted. Similarly, while analysis showed that studies that treated affect as a continuum (rather than separate entities) and that were published in China and France demonstrated lower effect sizes, the degrees of freedom were lower than four, hence this estimate could not be trusted. Non-significant predictors were omitted from the final model. See Table 2 for a summary of moderator analysis results.

## Association between daily positive affect and volume of alcohol consumed

A total of 50 effect sizes were extracted from 35 studies (6384 participants). Correlations between negative affect and consumption ranged from -.19 to .96. The pooled correlation

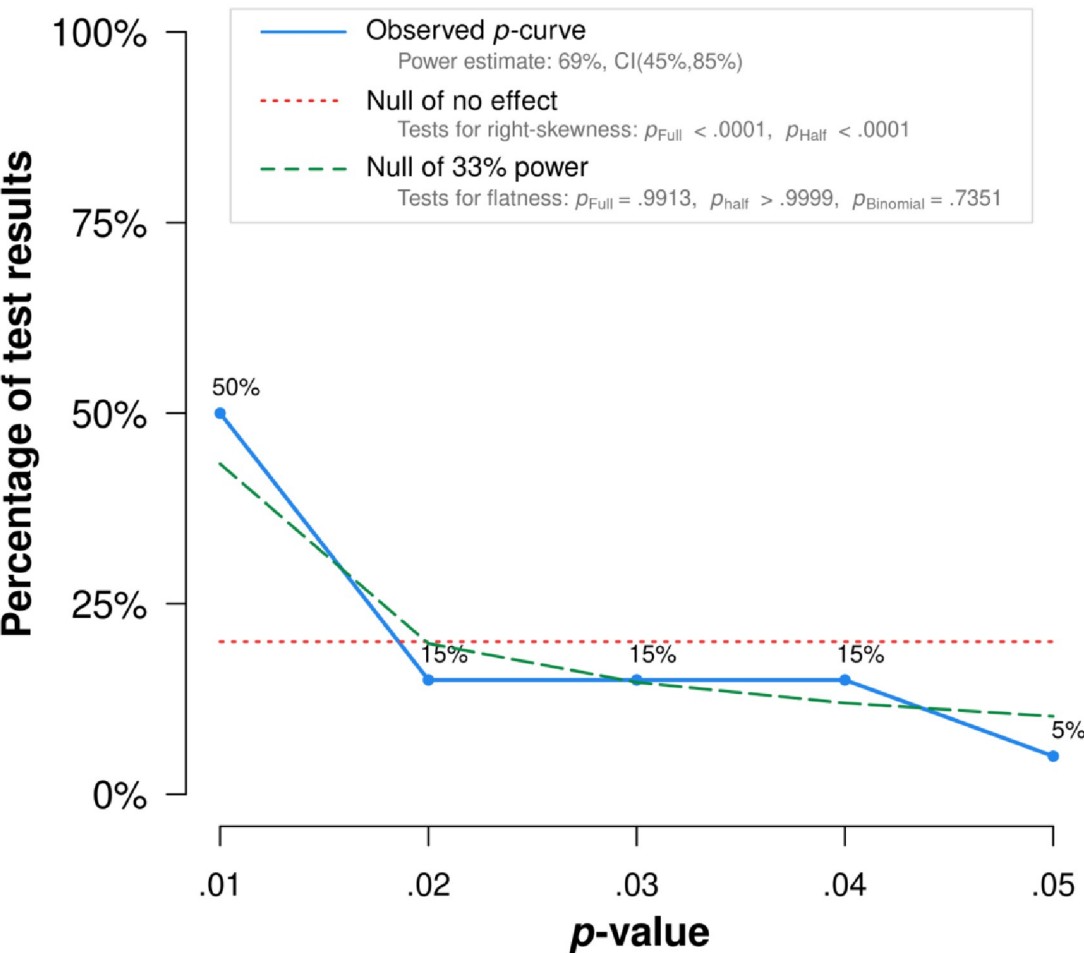

**Fig 2. P-curve plot for studies on negative affect and alcohol consumption.** Note: The observed *p*-curve includes 20 statistically significant (*p* < .05) results, of which 15 are *p* < .025. There were 107 additional results entered but excluded from *p*-curve because they were *p* > .05.

coefficient for our data was $r = .17$, 95% CI [.04, .30], $t$ (34) = 2.70, $p = .011$. As per $I^2$ and $\tau^2$ indexes, $I^2 = 98.29\%$, whereas $\tau^2 = .31$.

**Publication bias.** According to Egger's test, there was a publication bias ($t = 3.01$, $p = .006$). Selection model also demonstrated publication bias, $\chi^2$ (1) = 36.35, $p < .001$. After adjusting for publication bias, the relationship between negative affect and drinking volume was still positive and significant, $r = .52$, 95% CI [.35, .66], $p < .001$.

*P*-curve analysis indicated that evidential value is present, and that evidential value is not absent or inadequate (see Fig 4). This means that *P*-curve estimates that there is a "true" effect size underlying finding, and that the results are unlikely to be the product of publication bias and *p*-hacking alone. When correcting for selective reporting, the power of tests included into meta-analysis was 96% (see Fig 5).

**Meta-regression.** Several moderators were examined: year of publication, country, study quality, study design (laboratory vs field), alcohol measure, whether studies examined distinct emotions or averaged them, and whether study considered affective state to be a continuum or not. While measuring amount consumed in units was a significant predictor of higher effect sizes in the initial model, this was not significant anymore when the model was reduced. Year

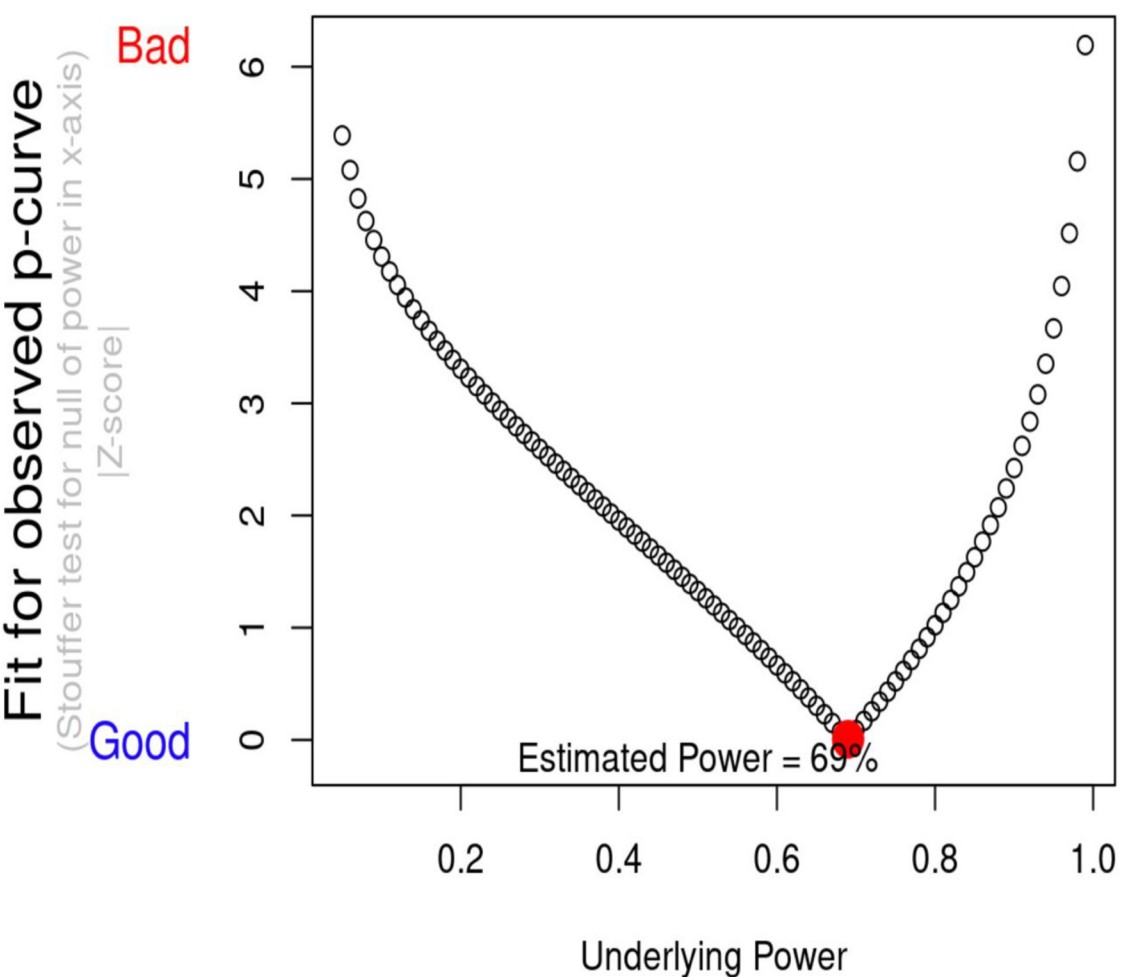

**Fig 3. Estimated power of meta-analysis on negative affect and drinking volume.**

**Table 2. Moderators of the relationship between negative affect and alcohol consumption volume.**

|  | Estimate | SE | t | df | 95% CI | *P* |
|---|---|---|---|---|---|---|
| Intercept | .12 | .12 | 1.01 | 1.34 | [-.71, .95] | .460 |
| Canada | .04 | .11 | .34 | 2.24 | [-.41, .48] | .761 |
| China | -.43 | .11 | -4.06 | 2.29 | [-.83, -.03] | .044 |
| France | -.46 | .11 | -4.35 | 2.29 | [-.86, -.06] | .038 |
| Netherlands | .07 | .11 | .66 | 2.17 | [-.37, .51] | .570 |
| UK | .08 | .11 | .77 | 2.29 | [-.32, .48] | .512 |
| USA | .11 | .10 | 1.02 | 1.11 | [-.94, 1.56] | .480 |
| Measuring affect as continuum | -.30 | .03 | -11.81 | 2.20 | [-.40, -.20] | .005 |
| Number of drinks as an alcohol measure | -.19 | .06 | -3.49 | 23.68 | [-.31, -.09] | .002 |
| Number of units as an alcohol measure | .46 | .04 | 10.99 | 2.28 | [.30, .63] | .005 |
| Number of sips as an alcohol measure | .10 | .06 | 1.67 | 14.47 | [-.03, .22] | .116 |

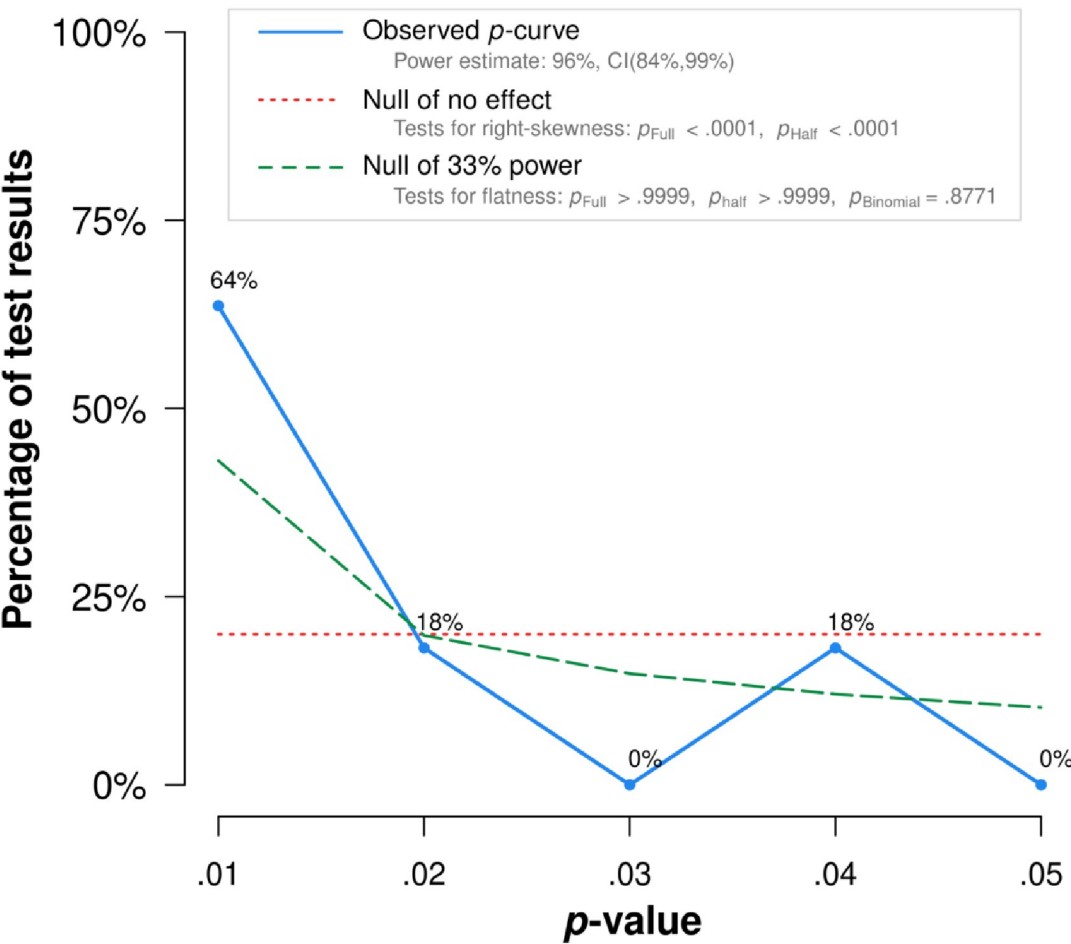

**Fig 4. *P*-curve plot for studies on positive affect and alcohol consumption.** Note: The observed *p*-curve includes 11 statistically significant (*p* < .05) results, of which 9 are *p* < .025. There were 39 additional results entered but excluded from *p*-curve because they were *p* > .05.

published was a significant moderator—with later years, effect sizes decreased, *t* (48) = -2.93, *p* = .005. None of the other moderaors were significant.

## Comparing the results of meta-analyses of negative and positive affect

Cohen's *q* statistics was calculated by comparing the results for negative affect (*r* = .09) and positive affect (*r* = .17). The obtained *q* value was .08, indicating that there is no significant difference between the effect sizes of two coefficients.

## Discussion

With the aim of examining the extent to which alcohol consumption can be explained theoretically by accounts which posit that people drink to enhance positive or to overcome negative affective states, meta-analyses of eligible non-clinical research outputs spanning 46 years were performed. Findings can be summarized as follows. First, both elevated negative and increased positive affect were associated with increased alcohol consumption volume, although the effect sizes were small. This may indicate that the mixed findings to date may be due to a

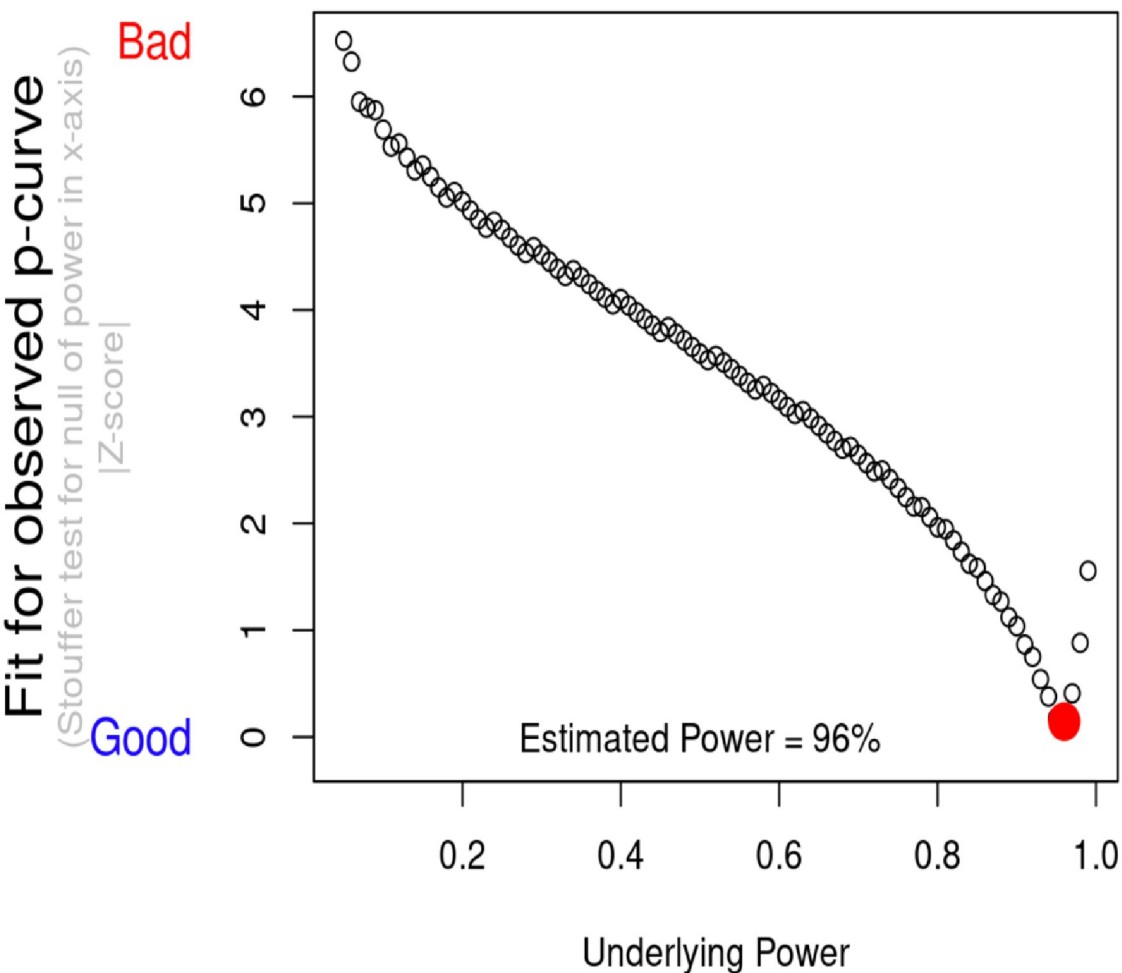

**Fig 5. Estimated power of meta-analysis on positive affect and drinking volume.**

predominance of underpowered individual studies in this field of research. Second, we did not find that affect measure used impacted the nature of the results observed within studies. Third, for negative affect, studies that used number of drinks as the alcohol consumption measure found lower effect sizes than research that used other metrics (i.e., number of units, amount in milliliters, number of sips, number of drinks). Other moderators were also examined in exploratory analyses (e.g., country), though the only significant moderator of effect sizes for positive affect was year published, pointing to a tendency for effect sizes to decline over time.

### The relationship between daily affect and alcohol consumption volume: Theoretical implications

The results of our meta-analysis indicate that both negative and positive affective states over the course of a day are associated with increased consumption volume in non-clinical populations. This temporal positive association is consistent with affect regulation models of alcohol consumption (self-medication hypothesis, [3]; tension-reduction theory, [11]; stress-response dampening theory, [12]; negative reinforcement model of alcohol use, [4]; positive reinforcement theory of alcohol use, [26]). Our analyses build on the meta-analysis by Bresin et al. [2],

which found that laboratory negative mood induction is associated with increased consumption, and extend this work in two ways. First, we demonstrate that the association between increased negative affect and alcohol consumption occurs in field as well as laboratory studies, thereby helping to overcome concerns regarding the ecological validity of laboratory-based work (although laboratory studies yielded higher effect sizes for negative affect). Second, we also found that positive affect, not previously considered meta-analytically, was also associated with elevated same day alcohol consumption.

The finding that both negative and positive affect are associated with increased alcohol consumption raises questions about whether it is necessary to retain both negative and positive reinforcement models of alcohol use, or whether a more parsimonious theoretical account of the mood-alcohol nexus may be possible. There are two reasons for entertaining this thought. First, the effect sizes for negative and positive affect were similar (.09 difference), and, when compared statistically using the $q$ denominator [79], were not significantly different from each other. Therefore, increases of affect (i.e., affect intensity, [142,143]) may play a more determinant role than affect valence (i.e., pleasantness) in explaining the relationship between mood and consumption. Second, as indicated by the moderator analysis, whether distinct emotions were examined did not appear to impact the affect-alcohol relationship. Our analyses, in this way, indicate that specific emotions are not differently associated with consumption, but the intensity of these emotions is. For example, it did not make a difference whether happiness or sadness were examined as predictors of alcohol consumption; rather how strongly happiness or sadness were experienced appeared to be important.

What emerges from these findings is that it may be useful to subsume existing affect regulation models, which posit that alcohol consumption is driven by a desire to alleviate or heighten particular affective states, with an account that emphasises affect intensity: it may be that particular affective states *per se* are less important in explaining increased alcohol use than the regulation of their intensity. This model would suggest that, on a given drinking day, alcohol consumption is likely to be elevated in individuals whose mood is more intense (or whose affect is less 'flat'). Such an approach does not contradict the notion that negative and positive affect are distinct entities. Instead, it asserts that the contents of emotions are of less importance in explaining alcohol consumption than the intensity with which these are experienced. The proposed account is consistent with the notion that people can experience negative and positive affect at the same time [57], and postulates that both may simultaneously shape alcohol consumption.

This way of thinking about drivers of alcohol consumption may have wider theoretical implications. For example, when considering the drinking motives literature [144,145]—which is also characterised by inconsistent and mixed finding (although see meta-analysis by [146])—it may be possible to think about distinct emotional drinking motive categories (enhancement and coping, [147]) in terms of 'affect regulation'. In this way, Littlefield et al. found that drinkers who consume alcohol for either coping or enhancement motives do not form two distinct groups [14], suggesting that these motives may be best viewed as dimensional variables that covary such that individuals who are high in one internal motive tend to be high in the other motive. This is not to say that negative and positive affect (or coping / enhancement motives) predict all forms of alcohol consumption in the same way. Nevertheless, based on the results of our meta-analyses, these factors may similarly be associated with daily drinking volume on a drinking day. Future research could fruitfully investigate whether affect intensity is associated with other variables of interest such as drinking onset or craving.

Methodologically, an affect intensity regulation hypothesis of alcohol consumption suggests that studies could focus on affect intensity instead of overall levels of affect. This could involve asking participants to report the extent to which they feel the intensity of their negative and

positive mood rather than asking them multiple questions about distinct emotions. Approaches requiring participants to only express the intensity of their overall negative and positive mood could lead to decreased participant burden and increased compliance in studies, which is particularly relevant to real-time research designs. While more research (in clinical populations) is required, current findings suggest that prevention and intervention efforts might usefully target overall levels of affect rather than focusing on affect valence. As such, providing individuals with alternative strategies for improving affect regulation may be of particular benefit.

## Moderator analyses

Moving on from possible theoretical implications of current findings, it is worthwhile to briefly consider significant moderators of the affect-alcohol consumption relationship. For negative affect, we found that studies that used number of drinks as an outcome tended to generate lower effect sizes than research that used other measures of drinking behavior such as the amount consumed in milliliters. For positive affect, year of publication was a significant moderator pointing to a tendency for effect sizes to decline over time. This is perhaps an indication that the field of alcohol research is not immune to the well-documented decline effect [148] whereby effect sizes can decrease over time for a variety of possible reasons that include false positive results, overestimation of effect sizes, under-specification of the conditions of the study, or genuinely decreasing effect sizes [149]. In the current context, it is also possible that advances in methodology and statistical analysis may have contributed to a more accurate effect size detection with the passage of time.

## Limitations and further research

The results of the current meta-analyses need to be considered in light of a number of limitations. First, the original correlation coefficients were not always available, and were extracted from standardised beta weights [65] which were converted from *d*, obtained from unstandardized beta weights [66], or *F*-values [67], using an online effect size calculator [Lenhard & Lenhard, Unpublished]. Furthermore, as *r* can only be obtained from standardised beta coefficient when it is between -.05 and .05, one study had to be excluded from analyses, while another study was omitted as it did not report any statistics from which effect sizes could be calculated. As such, the current meta-analysis for negative affect could not exhaustively represent all published data, although sensitivity analyses conducted indicated that there would have been no significant differences if it had been possible to use the excluded studies in the analyses. Second, while we separated the meta-analyses based on affect valence, we did not account for difference in affect arousal and how this could potentially impact the relationship between mood and consumption.

It is also important to note that our meta-analyses were concerned with drinking volume. That is, we examined whether intra-day affect influenced the amount of alcohol consumed on the day and future research could therefore usefully examine other variables of interest such as drinking onset, likelihood, blood alcohol concentration, or alcohol cravings. Similarly, further studies may consider examining the differences in the relationship between daily affect and heavy drinking (versus any drinking, as was examined in the present review). Moreover, this review only focused on intra-day consumption. While this allowed us to examine the association between state affect, further examination of trait affect (i.e., tendency to experience particular affective states) could help answer the question how longer-term affective states may be associated with alcohol consumption (e.g., [150]). Furthermore, the current meta-analyses focused on affective states in general without looking at potential differences between mood and emotional state. While this decision was made because there is an overlap between these

constructs and, for this reason, many studies use these terms interchangeably, further meta-analytical studies may wish to look at conceivably different effects of mood versus emotional state on alcohol consumption.

We also recommend that future research should routinely report the direct relationship between mood and consumption, and include correlation coefficients between all variables of interest. More generally, there is also a need for studies to be adequately powered and to conduct longitudinal investigations given the dominance of cross-sectional work in this area. As outlined, future research may also benefit from utilising affect intensity as a primary outcome variable as this could help reduce participant burden. Furthermore, since most of the studies on the topic were conducted in USA, additional research in other national contexts, which may differ with regards to the sociocultural positioning of alcohol [151], is advised.

## Conclusion

Overall, results of the present meta-analyses converge to suggest that both positive and negative affective states are associated with elevated daily alcohol consumption volume in non-clinical populations. While in apparent support of both positive and negative reinforcement models, present findings thereby suggest that greater insights into the relationship between mood and alcohol may be garnered through a more parsimonious focus on the intensity of emotional experiences (i.e., aggregate intensity of both negative and positive affect) rather than on valence. Consistent with idea that facets of positive and negative affect may be experienced simultaneously it therefore appears possible to posit an affect intensity regulation hypothesis. According to this, the intensity (rather than valence) of people's affective states on a given drinking day is associated with increased consumption of alcohol. While future research is required to test this theory, it is evident that much remains to be uncovered with regards to the mood-alcohol nexus, and that this endeavour will continue to exercise philosophers, writers, and scientists for some time to come.

## Supporting information

**S1 Checklist. PRISMA 2009 checklist.**
(DOC)

**S1 File. S1, S2—Unpussblished references.**
(DOCX)

## Author Contributions

**Conceptualization:** Anna Tovmasyan.

**Data curation:** Anna Tovmasyan, Rebecca L. Monk, Derek Heim.

**Formal analysis:** Anna Tovmasyan.

**Investigation:** Anna Tovmasyan.

**Methodology:** Anna Tovmasyan.

**Project administration:** Anna Tovmasyan.

**Resources:** Anna Tovmasyan.

**Software:** Anna Tovmasyan.

**Supervision:** Rebecca L. Monk, Derek Heim.

**Validation:** Anna Tovmasyan.

**Visualization:** Anna Tovmasyan.

**Writing – original draft:** Anna Tovmasyan.

**Writing – review & editing:** Anna Tovmasyan, Rebecca L. Monk, Derek Heim.

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
