## [Decision Letter · Decision Letter 0]

18 Nov 2021

PONE-D-21-30664Towards an affect intensity reinforcement hypothesis: A systematic review and meta-analyses of the relationship between affective states and alcohol consumptionPLOS ONE

Dear Dr. Tovmasyan,

Thank you for submitting your manuscript to PLOS ONE. After careful consideration, we feel that it has merit but does not fully meet PLOS ONE’s publication criteria as it currently stands. Therefore, we invite you to submit a revised version of the manuscript that addresses the points raised during the review process.

After review of the manuscript, both the reviewers and myself agree that there are major issues in the current version of the manuscript that need to be addressed/responded to by the authors. The major concerns are associated with the search strategy used by the authors that may have missed key literature for the meta-analysis. Also of concern is the fact that the authors indicate that they have followed the PRISMA guidelines and, for example, do not present a full electronic search strategy for at least one database, including any limits used, such that it could be repeated.  The search strategy and in which database, including limits used, should be indicated. Authors should strictly verify their adherence to the PRISMA guidelines in this manuscript.

We look forward to receiving your revised manuscript.

Kind regards,

Jose M. Moran

Academic Editor

PLOS ONE

2. We note that you have referenced (Lenhard W, Lenhard A. Computation of Effect Sizes [Internet]. Unpublished;)which has currently not yet been accepted for publication. Please remove this from your References and amend this to state in the body of your manuscript: (ie “Lenhard W, Lenhard. [Unpublished]”) as detailed online in our guide for authors

Reviewers' comments:

Reviewer's Responses to Questions

**Comments to the Author**

1. Is the manuscript technically sound, and do the data support the conclusions?

Reviewer #1: Yes

Reviewer #2: Yes

2. Has the statistical analysis been performed appropriately and rigorously? 

Reviewer #1: Yes

Reviewer #2: Yes

3. Have the authors made all data underlying the findings in their manuscript fully available?

Reviewer #1: Yes

Reviewer #2: Yes

4. Is the manuscript presented in an intelligible fashion and written in standard English?

Reviewer #1: Yes

Reviewer #2: Yes

5. Review Comments to the Author

Reviewer #1: The study attempted to clarify the pooled association between affective states and alcohol consumption by conducting a systematic review and meta-analysis. Overall, this study did a thorough and comprehensive discussion of previous literature and relevant theories. If the following issues can be addressed or clarified deeply, this study would be of good quality:

1. The authors stated that the operational definitions of mood and emotions in the Operational Definitions section, it is suggested that the operational definitions of alcohol consumption also be provided (page 8, line 189).

2. Although the three terms ‘mood’, ‘emotion’, and ‘feeling’ and ‘affect’ are often used interchangeably in the literature, there are many and significant debates regarding their similarities and differences in some aspects. The authors are encouraged to have a part discussing them in order to support the approach used: “... the terms ‘affect’, and ‘affective state’ are used in this review as umbrella terms for the experience of mood, emotion, or feeling”.

3. In the section of Literature Review, the authors used one of the commands for searching: ‘alcohol consumption’ rather than ‘alcohol’. This could lead to a loss of a certain amount of candidate studies for meta-analysis (page 9, line 225).

4. Also, in the section of Literature Review, the search term ‘feeling’ should be added as the authors mentioned ‘...as umbrella terms for the experience of mood, emotion, or feeling’ (page 9, line 226).

5. The meta-regression approach is used to work on “numeric moderator variables” such as year of publication. However, the study also applied this method, instead of the subgroup analysis approach, to examine the effects of categorical moderators (i.e., country and study design). Can the author clarify this?

6. Although no study with poor quality is included in this meta-analysis, running a moderator analysis of study quality has its great value.

7. In the ‘Meta-analysis – Analytical Strategy’ section, the authors mentioned ‘Several categorical moderators were examined ... alcohol consumption measure ...’. However, why the moderator analyses of such a variable (i.e., alcohol consumption measure) seem to be missing in Results.

8. In this meta-analysis, most of the included studies are from the USA; this may limit the extrapolation of the study’s result to some extent, even the related moderator analyses did not find an effect. It would be better that the study may have some discussion regarding this issue.

Reviewer #2: In this manuscript, Tovmasyan et al. performed meta-analyses for the relationship between affect and alcohol consumption. They found that the affect intensity, for both positive and negative affect, is associated with alcohol consumption. The study was well-conducted, the manuscript detailed the analysis approaches and methods, and the findings appear to be sound. The implications of the findings are likely to impact the field substantively.

Overall, there are no major issues with the manuscript. Minor issues are listed below:

1. Two separate figures were labeled as Figure 1: both the flowchart of the study selection process (line 329 – 372) and P-curve plot for studies on negative affect and alcohol consumption (line 435 – 438).

2. There were 149 references listed at the end of the manuscript. However, the text referred up to 157. The missing references need to be listed.

3. In the text, most of the time figures were referred to with a capital ‘F’. There were places where lower case ‘f’ was used (for example, line 431, 434, 469, 472). Please make it consistent.

4. Line 551: The reference number (142) should be after the word “literature”.

5. Line 614: Please add a comma after the word “consumption”.

6. PLOS authors have the option to publish the peer review history of their article (what does this mean?). If published, this will include your full peer review and any attached files.

Reviewer #1: No

Reviewer #2: No

---

## [Author Response · Author response to Decision Letter 0]

14 Dec 2021

Dear Dr Moran, 

Thank you for giving us the chance to revise this manuscript, and to both you and the reviewers for their constructive and helpful comments. Below we outline how we have responded to each reviewer comment. We have also uploaded a version of the manuscript with tracked changes, should this be desired by the reviewers. Having made these changes, we hope you will now find it acceptable for publication.

Best wishes,

Anna Tovmasyan 

**The major concerns are associated with the search strategy used by the authors that may have missed key literature for the meta-analysis. Also of concern is the fact that the authors indicate that they have followed the PRISMA guidelines and, for example, do not present a full electronic search strategy for at least one database, including any limits used, such that it could be repeated. The search strategy and in which database, including limits used, should be indicated. Authors should strictly verify their adherence to the PRISMA guidelines in this manuscript.

Thanks for this comment. The following has been added to the methods section in order to more fully outline our search strategy, as you suggest: “The search was conducted on 2nd March 2020. For PsychINFO, after the filters ‘empirical study’ and ‘quantitative study’ were applied, the search yielded 8285 articles for screening. For PsychARTICLES, when the same filters were applied, the search yielded three articles. For Science Direct, as wildcards “*” were not supported, the search terms were ("alcohol " OR "drinking behavior” OR “drinking behaviour”) AND ("mood" OR "emotions" OR “feelings” OR "affective states") NOT "disorders". After the filter ‘research articles’ was applied, the search yielded 2327 articles. For PubMed (3189 citations), SCOPUS (1201 citations), and JSTOR (367 citations), no filters were applied. The citations were loaded to RefWorks software, and the duplicates were removed. … To ensure that all relevant literature published at the time was covered, a supplementary search was conducted on 29th January 2021, which yielded 3 additional references”.

The search has also been re-run following these suggestions proposed by reviewers (specified in more detail below). This resulted in inclusion of one additional citation.

**When submitting your revision, we need you to address these additional requirements.

1. Please ensure that your manuscript meets PLOS ONE’s style requirements, including those for file naming. The PLOS ONE style templates can be found at

This has now been amended.

**2. We note that you have referenced (Lenhard W, Lenhard A. Computation of Effect Sizes [Internet]. Unpublished;)which has currently not yet been accepted for publication. Please remove this from your References and amend this to state in the body of your manuscript: (ie “Lenhard W, Lenhard. [Unpublished]”) as detailed online in our guide for authors

This has now been amended.

**3. Please include captions for your Supporting Information files at the end of your manuscript, and update any in-text citations to match accordingly. Please see our Supporting Information guidelines for more information: http://journals.plos.org/plosone/s/supporting-information.

This has now been amended.

Reviewers' comments:

**Reviewer #1: The study attempted to clarify the pooled association between affective states and alcohol consumption by conducting a systematic review and meta-analysis. Overall, this study did a thorough and comprehensive discussion of previous literature and relevant theories. If the following issues can be addressed or clarified deeply, this study would be of good quality:

1. The authors stated that the operational definitions of mood and emotions in the Operational Definitions section, it is suggested that the operational definitions of alcohol consumption also be provided (page 8, line 189).

Thank you for this helpful suggestion. The following has now been added to ‘Operational Definitions’ section: “Alcohol consumption is defined as ingesting any beverage containing ethanol.”

**2. Although the three terms ‘mood’, ‘emotion’, and ‘feeling’ and ‘affect’ are often used interchangeably in the literature, there are many and significant debates regarding their similarities and differences in some aspects. The authors are encouraged to have a part discussing them in order to support the approach used: “... the terms ‘affect’, and ‘affective state’ are used in this review as umbrella terms for the experience of mood, emotion, or feeling”.

Thank you for this valuable point. The following has now been added to ‘Operational Definitions’ section: “While it is possible that mood and emotions have different effects on alcohol consumption volume over longer periods of time, the current focus was on the effects of within a shorter timeframe, where the distinction between mood and emotions is arguably less important”. As the terms are often being used interchangeably, we decided to include all of them to make sure all the relevant literature is included. 

In the discussion, we also now address this point and suggest that further research on the issue is required: “Furthermore, the current meta-analyses focused on affective states in general without looking at potential differences between mood and emotional state. While this decision was made because there is an overlap between these constructs and, for this reason, many studies use these terms interchangeably, further meta-analytical studies may wish to look at conceivably different effects of mood versus emotional state on alcohol consumption”. 

**3. In the section of Literature Review, the authors used one of the commands for searching: ‘alcohol consumption’ rather than ‘alcohol’. This could lead to a loss of a certain amount of candidate studies for meta-analysis (page 9, line 225).

**4. Also, in the section of Literature Review, the search term ‘feeling’ should be added as the authors mentioned ‘...as umbrella terms for the experience of mood, emotion, or feeling’ (page 9, line 226).

Thank you for highlighting these important considerations. Given that the focus of the current paper was on the acute impact of affect on alcohol consumption, the search term ‘alcohol consumption’ was used rather than just ‘alcohol’. Given your suggestion, however, we did also carry out a new search using the broader search terms. Similarly, we felt that adding the term ‘feeling’ was unlikely result in additional relevant literature, as our review focused on quantitative research, where the scales tend to measure either mood (e.g., POMS), emotions (e.g., STAI), or affect in general (e.g., PANAS). On the other hand, feelings are more commonly measured by qualitative research studies, which were not included in the current study. That being said, it is important to question one’s (potentially biasing) assumptions and thank you for highlighting this. Accordingly, we carried out an additional search using this broader search term. When the search with updates terms has been carried out, this resulted in inclusion of one additional citation to our meta-analyses. 

**5. The meta-regression approach is used to work on “numeric moderator variables” such as year of publication. However, the study also applied this method, instead of the subgroup analysis approach, to examine the effects of categorical moderators (i.e., country and study design). Can the author clarify this?

Of course. We followed the approach outlined in the book ‘Doing meta-analysis with R’ (Harrer et al., 2021, p. 198), according to which “subgroup analysis is nothing else than a meta-regression with a categorical predictor”. We have added the following clarification to our analytical strategy section “The effects of categorical moderators (i.e., country and study design) were assessed using meta-regression approach, as suggested by Harrer et al. (77).” 

**6. Although no study with poor quality is included in this meta-analysis, running a moderator analysis of study quality has its great value.

We agree that this was an oversight. An analysis with study quality as a moderator has now been conducted and while study quality was not a significant moderator, this has now been added to the text.

**7. In the ‘Meta-analysis – Analytical Strategy’ section, the authors mentioned ‘Several categorical moderators were examined ... alcohol consumption measure ...’. However, why the moderator analyses of such a variable (i.e., alcohol consumption measure) seem to be missing in Results.

Thank you for highlighting this omission – the results of this moderator analysis have now been added to the manuscript.

**8. In this meta-analysis, most of the included studies are from the USA; this may limit the extrapolation of the study’s result to some extent, even the related moderator analyses did not find an effect. It would be better that the study may have some discussion regarding this issue.

This is a valuable point, which we now include in the discussion where we highlight the need for additional research in other national contexts, which may differ with regards to the sociocultural positioning of alcohol (p. 59).

**Reviewer #2: In this manuscript, Tovmasyan et al. performed meta-analyses for the relationship between affect and alcohol consumption. They found that the affect intensity, for both positive and negative affect, is associated with alcohol consumption. The study was well-conducted, the manuscript detailed the analysis approaches and methods, and the findings appear to be sound. The implications of the findings are likely to impact the field substantively.

Overall, there are no major issues with the manuscript. Minor issues are listed below:

***1. Two separate figures were labeled as Figure 1: both the flowchart of the study selection process (line 329 – 372) and P-curve plot for studies on negative affect and alcohol consumption (line 435 – 438).

Thank you for highlighting this – this has now been amended.

**2. There were 149 references listed at the end of the manuscript. However, the text referred up to 157. The missing references need to be listed.

Thank you, we appreciate your highlighting of this error – it seems to be stemming from incorrect numbering in the text rather than missing references. We have now thoroughly checked the manuscript to ensure all the references are in the correct place.

**3. In the text, most of the time figures were referred to with a capital ‘F’. There were places where lower case ‘f’ was used (for example, line 431, 434, 469, 472). Please make it consistent.

Thank you – this has now been amended

**4. Line 551: The reference number (142) should be after the word “literature”.

Thank you – this has now been amended

**5. Line 614: Please add a comma after the word “consumption”.

Thank you – this has now been amended.

Once again, we thank the reviewers and editors for their valuable feedback. This has enabled us to strengthen considerably the submission.

---

## [Decision Letter · Decision Letter 1]

4 Jan 2022

Towards an affect intensity regulation hypothesis: A systematic review and meta-analyses of the relationship between affective states and alcohol consumption

PONE-D-21-30664R1

Dear Dr. Tovmasyan,

We’re pleased to inform you that your manuscript has been judged scientifically suitable for publication and will be formally accepted for publication once it meets all outstanding technical requirements.

Kind regards,

Jose M. Moran

Section Editor

PLOS ONE

Additional Editor Comments (optional):

Reviewers' comments:

Reviewer's Responses to Questions

**Comments to the Author**

1. If the authors have adequately addressed your comments raised in a previous round of review and you feel that this manuscript is now acceptable for publication, you may indicate that here to bypass the “Comments to the Author” section, enter your conflict of interest statement in the “Confidential to Editor” section, and submit your "Accept" recommendation.

Reviewer #1: All comments have been addressed

Reviewer #2: All comments have been addressed

2. Is the manuscript technically sound, and do the data support the conclusions?

Reviewer #1: Yes

Reviewer #2: Yes

3. Has the statistical analysis been performed appropriately and rigorously? 

Reviewer #1: Yes

Reviewer #2: Yes

4. Have the authors made all data underlying the findings in their manuscript fully available?

Reviewer #1: Yes

Reviewer #2: Yes

5. Is the manuscript presented in an intelligible fashion and written in standard English?

Reviewer #1: Yes

Reviewer #2: Yes

6. Review Comments to the Author

Reviewer #1: Well written and studied. The authors have carefully and detailedly commented the remarks of the reviewers on an earlier version of the manuscript and have taken most of the remarks of the reviewers into consideration for the current version.

Reviewer #2: (No Response)

7. PLOS authors have the option to publish the peer review history of their article (what does this mean?). If published, this will include your full peer review and any attached files.

Reviewer #1: No

Reviewer #2: No

---

## [Editor Report · Acceptance letter]

20 Jan 2022

PONE-D-21-30664R1 

Towards an affect intensity regulation hypothesis: Systematic review and meta-analyses of the relationship between affective states and alcohol consumption 

Dear Dr. Tovmasyan:

I'm pleased to inform you that your manuscript has been deemed suitable for publication in PLOS ONE. Congratulations! Your manuscript is now with our production department. 

Kind regards, 

on behalf of

Dr. Jose M. Moran 

Section Editor

PLOS ONE